# WI3D: Weakly Incremental 3D Detection via Visual Prompts

## Abstract

Class-incremental 3D object detection demands a 3D detector to *locate* and *recognize* novel categories in a stream fashion, while not forgetting its previously learned knowledge. However, existing methods require delicate 3D annotations for learning novel categories, resulting in significant labeling cost. To this end, we explore a label-efficient approach called **W**eakly **I**ncremental **3**D object **D**etection (**WI3D**), which teaches a 3D detector to learn new object classes using cost-effective 2D visual prompts. For that, we propose a framework that infuses (i) class-agnostic pseudo label refinement module for high-quality 3D pseudo labels generation, (ii) cross-modal knowledge transfer for representation learning of novel classes, and (iii) reweighting knowledge distillation for preserving old class information. Extensive experiments under different incremental settings on both SUN-RGBD and ScanNet show that our approach learns well to detect novel classes while effectively preserving knowledge of base classes, and surpasses baseline approaches in WI3D scenarios.

## 1 Introduction

Existing 3D detectors (Qi et al., 2019; Misra et al., 2021; Rukhovich et al., 2022; Wang et al., 2022b) have achieved remarkable performance learning to detect predefined classes in a static 3D environment. However, novel-class objects will emerge when deploying existing methods to in-the-wild and dynamic environments. To generalize the model well to novel classes, a straightforward approach would be to combine existing datasets with novel-class objects and train the model from scratch. However, this approach becomes impractical when frequent updates are necessary, as training on the entire dataset would be time-consuming (Cermelli et al., 2022). Meanwhile, fine-tuning the detector on novel-class samples alone will typically lead to catastrophically forgetting base classes, which is caused by changes of model parameters to accommodate new samples without accessing previous samples. Recently, incremental learning, which studies how to incorporate novel classes by training only on novel-class samples while preventing catastrophic forgetting issues, becomes eminent in various 2D and 3D vision tasks (PourKeshavarzi et al., 2021; Wang et al., 2022a; Zhao & Lee, 2022; Yang et al., 2023).

Prior works (Zhao & Lee, 2022; Zhao et al., 2022; Liang et al., 2023) have made initial attempts in the field of class-incremental 3D object detection using delicate 3D annotation for novel-class objects. However, acquiring a large number of fully-labeled point cloud data is prohibitively expensive due to the difficulty of both 3D data collection and annotation (Ren et al., 2021). Inspired by the human visual system that excels at learning new 3D concepts through 2D images, we propose to incrementally introduce novel concepts to a 3D detector with the visual prompts generated from a cost-free 2D teacher other than revisiting 3D annotations for both base and novel classes as shown in Fig. 1. We term this new task as *Weakly Incremental 3D object Detection* (WI3D), which incrementally updates the model without any manual annotation for the novel classes. To our best knowledge, we are the *first* attempt to address WI3D, an unexplored yet important problem.

WI3D has two major challenges: **1)** how to introduce novel classes to a 3D detector through 2D visual prompts incrementally, and **2)** how to retain base classes knowledge without revisiting any 3D annotations. Recent studies (Lu et al., 2023; Peng et al., 2022) have made great initial attempts to directly generate 3D pseudo labels from 2D predictions. However, these approaches could not sufficiently address the issue of the noise within the pseudo labels. The existence of noisy, inaccurate,

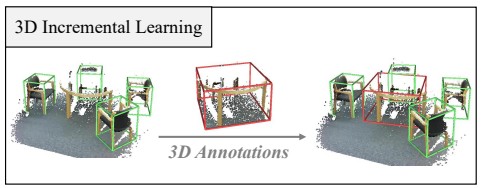 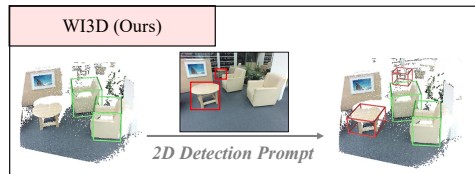

**Previous Class-Incremental 3D Object Detection**      **Weakly Incremental 3D Object Detection**

Figure 1: **Illustration of previous class-incremental 3D object detection (left) and WI3D (right).** Previous class-incremental 3D object detection methods rely heavily on the continual provisions of human annotations on the point cloud for novel classes. In contrast, we explore WI3D, a new task that introduces novel concepts to a 3D detector through 2D images to reduce the heavy cost of annotating the point cloud.

and incomplete pseudo labels severely deteriorates the detection performance in WI3D. Furthermore, the widely adopted knowledge distillation techniques (Zhao & Lee, 2022; Zhao et al., 2022) treat different Regions-of-Interests equally, leading to the failure to learn discriminative region features among the sparse and cluttered point cloud scenes.

To address the above issues, we propose a novel framework for WI3D with both intra- and inter-modal teachers, where the intra-modal teacher is a base 3D detector, and the inter-modality teacher is a 2D foundational model. Our framework is supervised by 1) the pseudo labels generated by both teachers and 2) concept representation learning in feature space. To obtain accurate pseudo labels, we propose a class-agnostic pseudo-label refinement module by learning the general and intrinsic latent relationship between the bounding boxes and the corresponding point cloud. In addition to incrementally teaching the current detector to localize novel objects in an explicit way, we also leverage an implicit way of supervision by learning in feature space. We propose an auxiliary cross-modal knowledge transfer for WI3D, which leverages bipartite matching to transfer color and texture-aware information from the visual prompts to enhance the 3D object representation. Finally, we explore a reweighting knowledge distillation approach that can discern and select the valuable knowledge of the existing classes, leading to further improvements in performance.

To summarize, our contributions are listed as the following:

- We introduce Weakly Incremental 3D object Detection (WI3D), a novel task that generalizes the base 3D detectors well to novel classes via cost-effective visual prompts only.
- We analyze the challenges in WI3D and propose a robust and effective framework for WI3D, which contains a class-agnostic pseudo label refinement module for high-quality pseudo labels generation and learning concept representation learning in feature space for both base and novel classes.
- Extensive experiments on two benchmark datasets, SUN RGB-D and ScanNet, illustrate the effectiveness of our methods under the low-cost setting of WI3D scenarios.

## 2 RELATED WORK

We first briefly review existing methods for class-incremental detection in 2D and 3D. Then, we introduce work on weakly-supervised 3D detection and the design of existing 3D object detectors.

**Class-Incremental Detection** explores the task of incrementally learning and detecting new classes over time while preserving the original capabilities of the detector as much as possible. Peng et al. (2020); Yang et al. (2022); Feng et al. (2022); Liu et al. (2023b) have made great efforts to class-incremental image object detection. Concurrently, several attempts for class-incremental 3D object detection are proposed. SDCoT (Zhao & Lee, 2022) proposes a static-dynamic co-teaching method for class-incremental 3D object detection. DA-CIL (Zhao et al., 2022) proposes a 3D domain adaptive class-incremental object detection framework with a dual-domain copypaste augmentation method to adapt the domain gradually. Recent work I3DOD (Liang et al., 2023) proposes a task-shared prompts mechanism to learn the matching relationships between the object localization information and category semantic information for class-incremental 3D object detection. In this paper, we

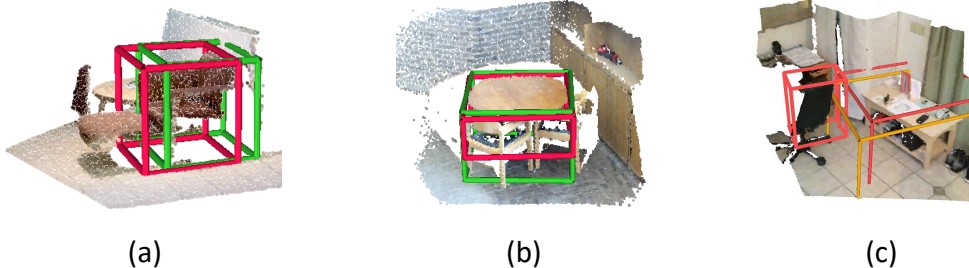

(a)                                        (b)                                        (c)

Figure 2: **The challenge of generating accurate 3D pseudo-labels from 2D visual prompts.** (1) *Projection Migration* occurs due to the background pixels within a 2D bounding box, causing the displacement of the 3D bounding box position compared with ground-truth label. (2) *Scale Ambiguity.* Due to the sparse representation of object surfaces, the pseudo labels generated cannot encompass the entire table legs. (3) *Overlapped Boxes* when aggregating pseudo labels from multi-view images.

explore a new paradigm, WI3D, to study how 2D knowledge enables a 3D detector to learn novel objects continually, without the reach for labor-consuming 3D annotations for the novel classes.

**Weakly-supervised 3D detection** studies a way to train a 3D detector without 3D instance annotation, such as object center annotations (Xu et al., 2022), scene-level class labels (Ren et al., 2021) and so on. The recent work, OV-3DET (Lu et al., 2023) introduces open-vocabulary 3D object detection, which directly utilizes a pre-trained 2D model to generate pseudo labels for a 3D detector. However, OV-3DET mainly focuses on the ability to associate each 3D instance with an appropriate text prompt, which can't handle the problems of *incremental localization* and *incremental semantic recognition* of emerging objects in the scene. In addition, how to acquire *accurate* 3D pseudo labels from 2D predictions remains unexplored in OV-3DET. In this paper, we study the potential of utilizing 2D visual prompts in weakly incremental 3D detection scenario by learning from denoised pseudo labels and regional concept representation.

**3D Object Detectors** requires a model to localize objects of interest from a 3D scene input. (Qi et al., 2019; Zhang et al., 2020; Misra et al., 2021; Liu et al., 2021) manage to operate directly on the point clouds for 3D object detection. VoteNet (Qi et al., 2019) and H3DNet (Zhang et al., 2020) achieve end-to-end 3D object detection based on sampling, grouping, and voting operators designed especially for point clouds. 3DETR (Misra et al., 2021) and GroupFree3D (Liu et al., 2021) extend the transformer (Vaswani et al., 2017) architecture to 3D object detection. In our paper, we adopt the modified VoteNet (Zhao & Lee, 2022) as our detection backbone, and explore how to extend a base 3D detector with the ability to detect objects of novel classes through 2D visual prompts.

## 3 METHODOLOGY

In this section, we first provide the task setting of WI3D and present the noise of 3D pseudo labels directly generated from 2D predictions in Sec. 3.1 and Fig. 2. Then, we provide the overview of our framework for WI3D in Sec. 3.2, which supervises the 3D detector with both the denoised pseudo labels (Sec. 3.3) and representation learning in feature space(Sec. 3.4).

### 3.1 PROBLEM DEFINITION

**Task Definition.** Given a base 3D detector capable of localizing and recognizing the base category set $C_{base}$ from a point cloud, WI3D extends its capacity to detecting a larger category set $\mathcal{C}_{all} = \mathcal{C}_{base} \cup \mathcal{C}_{novel}$ with only visual prompts for $C_{novel}$ from off-the-shelf 2D models. Here, we assert that each 3D scene is constructed from RGB-D images.

**Coarse Pseudo Labels Generation.** To generate novel-class pseudo labels for $\mathcal{S}^{3D}$ without point-level annotation, we leverage visual prompts generated by a cost-free 2D teacher $\mathcal{T}^{2D}$. Despite the one-to-one correspondence between points and pixels for each scan collected by RGB-D cameras, it is hard to directly localize 3D object and estimate a tight 3D bounding box with only a 2D one. Thus, we adopt a simple way to generate coarse 3D pseudo labels from 2D predictions, as mentioned in (Peng et al., 2022).

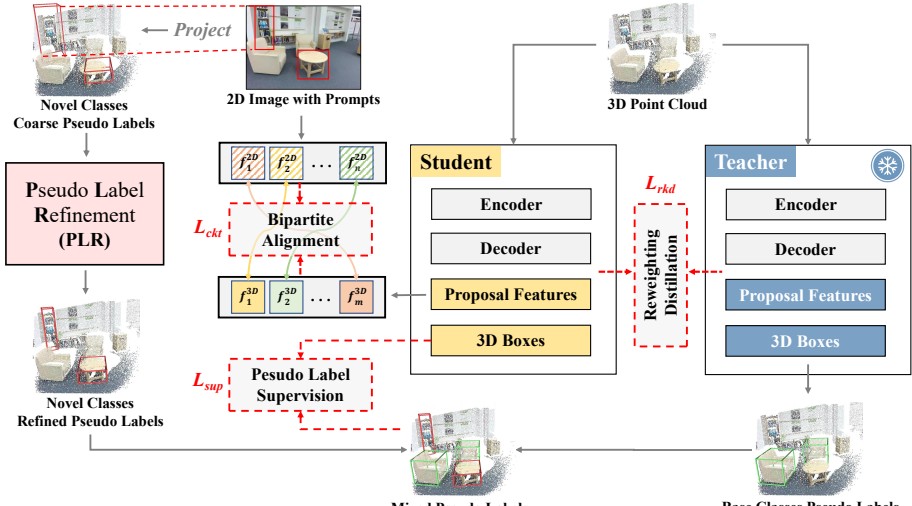

Figure 3: **Pipeline of our proposed WI3D.** We train a 3D student detector $\mathcal{S}^{3D}$ on (1) 3D pseudo labels and (2) visual concept representations generated by both the inter-modal teacher $\mathcal{T}^{2D}$, and intra-modal teacher $\mathcal{T}^{3D}$. To be specific, the 3D pseudo labels serve as direct supervision for $\mathcal{S}^{3D}$, and are generated by denoising and mixing the predictions of $\mathcal{T}^{2D}$ and $\mathcal{T}^{3D}$. Concurrently, the visual concept representation learning includes the cross-modal regional feature alignment for novel class and reweighting knowledge distillation for base classes. (Color is used for visualization only.)

**Noise analysis.** Specifically, projecting 2D bounding boxes into 3D space leads to the following noise: *Projection Migration*: As shown in Fig. 2 (a), background pixels within 2D bounding boxes lead to the displacement of the 3D bounding box positions. (2) *Scale Ambiguity*: As shown in Fig. 2 (b), the scale ambiguity problem is caused by sparse points captured from the surface of an object, leading to untight dimension estimation for the pseudo label. (3) *Overlapped Boxes*: As shown in Figure Fig. 2 (c), duplicated estimations on the same instance will occur when fusing multi-frame predictions (e.g. the red and yellow pseudo labels represent the predicted results from two consecutive frames, respectively).

## 3.2 PIPELINE OVERVIEW

Our pipeline is initialized with a base 3D detector $\mathcal{T}^{3D}$, which is capable of detecting $\mathcal{C}_{base}$. As is shown in Fig. 3, in order to train a 3D detector $\mathcal{S}^{3D}$ for both incrementally detecting *novel* classes and retaining *base* knowledge without any reach of 3D annotations for either base or novel classes, we seek supervision from both pseudo labels and feature space. More accurate 3D pseudo labels become within reach for $\mathcal{S}^{3D}$ by further adopting the proposed pseudo label refinement module in Sec. 3.3. Additionally, cross-modal knowledge transfer and reweighting knowledge distillation in Sec. 3.4 serve as great feature-level supervision for class-incremental learning.

## 3.3 PSEUDO LABEL REFINEMENT

**Class-Agnostic Pseudo Label Refinement.** To generate tight 3D pseudo labels from 2D predictions, we propose the class-agnostic **P**seudo-label **ReF**inement (PRF) module (Fig. 4). To be specific, PRF includes a light-weight PointNet (Qi et al., 2017) encoder $\mathcal{PN}$ to encode contextual information from point cloud $\tilde{p}$, and a MLP based box encoder $MLP_{box}$ to encode positional information of the 3D bounding box $\tilde{b}^{3D}$, and a decoder $\mathcal{D}$ to obtain the box offset.

$$\hat{b}^{3D} = \tilde{b}^{3D} + PRF\left(\tilde{p}, \tilde{b}^{3D}\right) = \tilde{b}^{3D} + \mathcal{D}\left(\left[\mathcal{PN}\left(\tilde{p}\right); MLP_{box}\left(\tilde{b}^{3D}\right)\right]\right). \qquad (1)$$

Here, $[\cdot; \cdot]$ is the concatenation operation. Since PRF is class agnostic, we can train it on base classes and adopt it directly to novel classes during class-incremental training.

Figure 4: **The design of Pseudo Label Refinement(PRF).** Our proposed PRF first encodes the coarse 3D box coordinates and the normalized point cloud. Then, these two feature are concatenated and used for box coordinate refinement. (Color is used for visualization only.)

We decouple and refine the parameters of each 3D box, including its location, dimensions, and orientation, and solve the problem caused by *Projection migration* and *Scale ambiguity*. However, the aforementioned approach does not address the issue of *Box overlap*. In order to address this, we propose a method to determine the validity of each 3D pseudo label by incorporating an additional **B**inary **C**lassification **H**eader (**BCH**). The inputs to our BCH are the contextual feature from point cloud and box-aware embedding and the outputs of BCH are the binary probability. We use the Hungarian algorithm (Kuhn, 1955) to match each annotation of the base class with an input pseudo label, and the coarse pseudo-labels that do not match any annotation will be masked by BCH. During inference, the bounding box is considered valid only when the probability of presence exceeds the probability of absence.

### 3.4 CONCEPT REPRESENTATION LEARNING

Beyond the learnable denoising module for generating high-quality 3D pseudo boxes, we introduce auxiliary objectives to enhance the student's robust representation learning capability in an implicit way.

**Cross-modal Knowledge Transfer.** Because of the occlusion of objects in 2D images, background objects are often included when extracting visual feature of a specific region, which will further lead to confusing feature representations when serving as the source of supervision. To this end, we propose **C**ross-modal **K**nowledge **T**ransfer (CKT) to help the model learn robust feature representations. Inspired by (Lin et al., 2022), we frame the cross-modal alignment assignment as a bipartite matching problem. In practice, we project the box estimation generated by $\mathcal{S}^{3D}$ onto the corresponding image and build the matching matrix by calculating IoU between the projected 3D box with the 2D predictions generated by $\mathcal{T}^{2D}$ The cost function for bipartite matching can be formulated as follows:

$$\begin{cases} min \sum_i \sum_j m_{ij} * IoU(Project(B_i^{3D}), B_j^{2D}) \\ \sum_i m_{ij} = 1 \end{cases} \tag{2}$$

where $m_{ij} \in 0, 1$ indicates whether it matches, and $IoU$ represents computing the intersection over union. Then a pretrained image region encoder $\mathcal{E}^{2D}$ is used to extract the features of the novel-class instance $R_j^{2D}$ from image, denoted as:

$$F_j^{2D} = \mathcal{E}^{2D}(R_j^{2D}) \tag{3}$$

For 3D proposal $B_m^{3D}$ paired with corresponding $B_n^{2D}$, we feed the proposals' features $F_m^{3D'}$ in $B_m^{3D}$ into an MLP-based projection head $\mathcal{H}^{3D}$, to encode the 3D proposal features into the same feature space of $F_n^{2D}$, denoted as $F_m^{3D} = \mathcal{H}^{3D}(F_m^{3D'})$. Finally, we design a dynamic instance-level knowledge transferring loss based on the negative cosine similarity (Chen & He, 2020), assigning different weights to different instance samples:

$$\mathcal{L}_{ckt} = - \sum_{m \in M, n \in N} \frac{F_m^{3D}}{\|F^{3D}\|_2} * \frac{F_n^{2D}}{\|F^{2D}\|_2} \tag{4}$$

**Intra-modal Base Knowledge Distillation.** To alleviate forgetting issues, existing works (Zhao & Lee, 2022; Zhao et al., 2022) use knowledge distillation Hinton et al. (2015) to preserve learned

knowledge. However, previous work usually directly utilizes all the predicted responses and treat knowledge equally, fails to capture discriminative region features in sparse and cluttered point cloud scenes. Here, we argue that not all features in the old model must be distilled, and we propose an **R**eweighting **K**nowledge **D**istillation scheme (RKD) that focuses on distilling the region with more considerable influence. The distillation loss $\mathcal{L}_{rkd}$ can be computed as:

$$\mathcal{L}_{rkd} = \frac{1}{K}\left[\sum_{i \in \Phi_B} \alpha_i(||F_i^S - F_i^T||_2 + ||l_i^S - l_i^T||_2)\right] \tag{5}$$

where $\Phi_B$ is the set of indices of base-class proposals and $K$ is the total number of proposals; $F_i$ and $l_i$ are the features and classification logits of $i_{th}$ proposal respectively; $\alpha_i$ is reweighting modulation factor obtained based on the proposal objectness $o_i$, denoted $\alpha_i = \frac{e^{o_i}}{\sum_{i=1}^{K} e^{o_i}}$; the superscripts $S$ and $T$ represent the features from the student and teacher model.

### 3.5 TRAINING OBJECTIVES

**Base training.** We train the modified VoteNet (Zhao & Lee, 2022) on base class annotations with the detection loss $\mathcal{L}_{det}$ (Qi et al., 2019), which is defined as

$$\mathcal{L}_{det} = \alpha_1\mathcal{L}_{vote} + \alpha_2\mathcal{L}_{obj} + \alpha_3\mathcal{L}_{box} + \alpha_4\mathcal{L}_{sem-cls}. \tag{6}$$

Here, $\alpha_1, \alpha_2, \alpha_3, \alpha_4$ are set as $1, 0.5, 1, 0.2$, and $\mathcal{L}_{vote}, \mathcal{L}_{obj}, \mathcal{L}_{box}, \mathcal{L}_{sem-cls}$ stands for vote regression, proposal objectness classification, box regression, and proposal semantic classification respectively. Note that we also train PRF on $C_{base}$ in this stage, where $\mathcal{L}_{PRF} = \mathcal{L}_{box}$.

**Weakly Incremental Learning.** The supervision of the WI3D comes in two folds: explicit detection training on the pseudo labels generated by $\mathcal{T}^{2D}$ and $\mathcal{T}^{3D}$ with $\mathcal{L}_{det}$, and the feature-space learning with instance-level knowledge transfer $\mathcal{L}_{ckt}$ and knowledge distillation of base classes $\mathcal{L}_{rkd}$. The loss function can be defined as

$$\mathcal{L} = \beta_1\mathcal{L}_{det} + \beta_2\mathcal{L}_{ckt} + \beta_3\mathcal{L}_{rkd}. \tag{7}$$

Here, $\beta_1, \beta_2, \beta_3$ are set as $1, 10, 5$ heuristically.

## 4 EXPERIMENTS

We first introduce the datasets, metrics, and implementation details for weakly incremental 3D object detection in Sec. 4.1. Then, we compare our methods with different baseline approaches and prior arts in Sec. 4.2. Afterward, we take out ablation studies to study the effectiveness of the proposed components in Sec. 4.3. Finally, we showcase some visualization results in Sec. 4.4.

### 4.1 DATASETS, METRICS, AND IMPLEMENTATION DETAILS

**Datasets.** Following previous works on class-incremental 3D detection (Zhao & Lee, 2022; Zhao et al., 2022; Liang et al., 2023), we conduct experiments on two widely used datasets, SUN RGB-D (Song et al., 2015) and ScanNet (Dai et al., 2017). SUN-RGBD consists of 10,335 single-view RGB-D scans, where 5,285 are used for training, and 5,050 are for validation. Each scan is annotated with rotated 3D boxes. ScanNet includes 1,201 training samples and 312 validation samples reconstructed from RGB-D sequences. We split the full category set into two non-overlapped subsets into $C_{base}$ and $C_{novel}$ according to (Zhao & Lee, 2022).

**Metrics.** To compare the performance of different approaches under incremental settings, we adopt $mAP_{base}$, $mAP_{novel}$, and $mAP_{all}$ as abbreviations for **m**ean **A**verage **P**recision (mAP) under an IoU threshold of 0.25 for base classes, novel classes, and overall performance, respectively.

**Implementation Details.** The input of our model is a point cloud $P \in \mathbb{R}^{N \times 3}$ representing a 3D scene, where $N$ is set as 20,000 and 40,000 respectively for SUN RGB-D and ScanNet. Following (Zhao & Lee, 2022), the base training lasts for 150 epochs using an Adam optimizer (Kingma & Ba, 2014) with a batch size of 8, and a learning rate of $10^{-3}$ decaying to $10^{-4}$ and $10^{-5}$ at the $80th$ and $120th$ epoch respectively. During weakly incremental learning, we copy weights from $\mathcal{T}^{3D}$ to initialize the student model $\mathcal{S}^{3D}$, and optimize $\mathcal{S}^{3D}$ under the supervision of both refined pseudo labels and feature space. During both training stages, we evaluate $\mathcal{S}^{3D}$ every 10 epochs. All experiments mentioned above are conducted on a single RTX3090 GPU.

| Method | $\lvert\mathcal{C}_{novel}\rvert = 3$ | | | $\lvert\mathcal{C}_{novel}\rvert = 5$ | | | $\lvert\mathcal{C}_{novel}\rvert = 7$ | | |
|---|---|---|---|---|---|---|---|---|---|
| | $\text{mAP}_{base}\uparrow$ | $\text{mAP}_{novel}\uparrow$ | $\text{mAP}_{all}\uparrow$ | $\text{mAP}_{base}\uparrow$ | $\text{mAP}_{novel}\uparrow$ | $\text{mAP}_{all}\uparrow$ | $\text{mAP}_{base}\uparrow$ | $\text{mAP}_{novel}\uparrow$ | $\text{mAP}_{all}\uparrow$ |
| base-training | 53.84 | - | - | 58.54 | - | - | 50.88 | - | - |
| fine-tuning | 1.02 | 35.41 | 11.34 | 1.11 | 32.98 | 17.05 | 0.13 | 27.25 | 19.12 |
| freeze-and-add | 53.05 | 9.99 | 40.13 | 56.29 | 5.99 | 31.21 | 47.11 | 1.95 | 15.50 |
| SDCoT | 39.64 | 45.38 | 41.36 | 49.27 | 34.08 | 41.68 | 49.75 | 25.83 | 33.00 |
| *Ours:* | | | | | | | | | |
| WI3D | 42.70 | **50.71** | **45.10** | 51.52 | **41.65** | **46.58** | 51.26 | **32.04** | **37.81** |
| *Upper Bounds:* | | | | | | | | | |
| 3D-GT | 44.82 | 67.69 | 51.68 | 54.27 | 58.89 | 56.58 | 55.10 | 56.73 | 56.24 |

Table 1: **Weakly incremental 3D object detection (mAP@0.25) on SUN RGB-D validation set.** All methods listed are first trained on base classes $\lvert C_{base}\rvert = 10 - \lvert C_{novel}\rvert$ before incremental learning novel classes $\lvert C_{novel}\rvert$. $\uparrow$ means the higher, the better.

## 4.2 COMPARISON WITH EXISTING METHODS

We construct several baseline methods to study this task: 1) ***Base-training*** directly train the 3D detector on base classes. 2) ***Fine-tuning*** tune the whole model (except the base classifier) and a new classifier for the $C_{novel}$. 3) ***Freeze-and-add*** freeze the backbone, followed by adding a new classification head and train only the new head on novel classes. Additionally, we modify the training of ***SDCoT*** (Zhao & Lee, 2022) to fit our weakly incremental learning setting. For a fair comparison, all the training settings, e.g., learning rate, optimizer, batch size, etc., are the same for all experiments.

To make thorough evaluations, we compare our method with all the above mentioned methods under different class-incremental settings on SUN RGB-D (Tab. 1) and ScanNet (Tab. 2). One shall notice that under different settings, the baseline methods either lead to catastrophic forgetting or failure to learn novel concepts. For instance, when we evaluate the methods on SUN RGB-D with $\lvert\mathcal{C}_{novel}\rvert = 5$, ***fine-tuning*** only achieves 1.11 $\text{mAP}_{base}$, and ***freeze-and-add*** achieves 5.99 $\text{mAP}_{novel}$. The former suffers from severe catastrophic forgetting on base classes, while the latter cannot learn new classes effectively. Additionally, it can be shown that our method can also surpass ***SDCoT***, which achieves 49.27% $\text{mAP}_{base}$, 34.08% $\text{mAP}_{novel}$ and 41.68% $\text{mAP}_{all}$ when $\lvert\mathcal{C}_{novel}\rvert = 5$, while our framework achieves 51.52% $\text{mAP}_{base}$, 41.65% $\text{mAP}_{novel}$(+7.57%), 46.58% $\text{mAP}_{all}$(+4.9%) under the same task setting on SUN RGB-D dataset. Compared to SDCoT, which experiences significant performance degradation when introducing novel classes, our method achieves a balance between base and novel classes across different class-incremental settings. These phenomena are prevalent and can be observed through experiments conducted on both datasets.

| Method | $\lvert\mathcal{C}_{novel}\rvert = 6$ | | | $\lvert\mathcal{C}_{novel}\rvert = 9$ | | | $\lvert\mathcal{C}_{novel}\rvert = 12$ | | |
|---|---|---|---|---|---|---|---|---|---|
| | $\text{mAP}_{base}\uparrow$ | $\text{mAP}_{novel}\uparrow$ | $\text{mAP}_{all}\uparrow$ | $\text{mAP}_{base}\uparrow$ | $\text{mAP}_{novel}\uparrow$ | $\text{mAP}_{all}\uparrow$ | $\text{mAP}_{base}\uparrow$ | $\text{mAP}_{novel}\uparrow$ | $\text{mAP}_{all}\uparrow$ |
| base-training | 51.01 | - | - | 58.37 | - | - | 64.70 | - | - |
| fine-tuning | 1.66 | 27.42 | 10.24 | 2.42 | 20.72 | 11.57 | 3.96 | 17.32 | 12.87 |
| freeze-and-add | 50.33 | 1.96 | 34.21 | 58.10 | 2.08 | 30.09 | 63.30 | 1.56 | 22.14 |
| SDCoT | 38.97 | 23.45 | 33.80 | 47.46 | 20.07 | 33.77 | 51.99 | 16.83 | 28.55 |
| *Ours:* | | | | | | | | | |
| WI3D | 41.26 | **30.17** | **37.56** | 49.61 | **29.82** | **39.72** | 52.94 | **27.37** | **35.89** |
| *Upper Bounds:* | | | | | | | | | |
| 3D-GT | 52.85 | 61.31 | 55.67 | 59.40 | 51.73 | 55.56 | 63.59 | 51.40 | 55.46 |

Table 2: **Weakly incremental 3D object detection (mAP@0.25) on ScanNet validation set.** All methods listed are first trained on base classes $\lvert C_{base}\rvert = 18 - \lvert C_{novel}\rvert$ before incremental learning novel classes $\lvert C_{novel}\rvert$. $\uparrow$ means the higher, the better.

## 4.3 ABLATION STUDY AND ANALYSIS

In this section, we organize ablation studies to study the effectiveness of the proposed components. Without further specification, the following experiments are conducted on SUN RGB-D under the $\lvert\mathcal{C}_{novel}\rvert = 5$ setting.

**Effectiveness of Pseudo Label Refinement (PRF).** To make a better comparison, we include several baseline methods, including directly training with the coarse 3D pseudo labels ("-"), **N**on-**M**aximum **S**uppression (Neubeck & Van Gool, 2006) ("NMS"), and PRF without the Binary Classification Head ("PRF w/o BCH") in Tab. 3. It can be seen that the full model of our proposed PRF efficiently improves the detection performance of novel classes (+3.25% $\text{mAP}_{novel}$). Since NMS is initially designed to drop duplicated box estimations, it cannot handle the challenge of noisy 3D pseudo labels

generated from 2D box estimations well. Additionally, BCH can efficiently select pseudo-labels with higher quality, and further improve the detection performance ($+0.96\%$ mAP$_{novel}$).

**The Input of Pseudo Label Refinement (PRF).** In Tab. 4, we investigate the input design of the PRF module. We notice that using either the coarse 3D pseudo boxes' spatial coordinates or the contextual information from the input point cloud will severely downgrade the detection performance, since either input is insufficient to provide adequate information to refine the coarse 3D pseudo boxes. Whereas, the model achieves the optimal performance when both are used as the input.

| Pseudo label denosing | mAP$_{base}$ ↑ | mAP$_{novel}$ ↑ | mAP$_{all}$ ↑ |
|---|---|---|---|
| - | 50.58 | 37.44 | 44.01 |
| NMS | 50.78 | 37.35 | 44.07 |
| PRF w/o BCH | 51.20 | 40.69 | 45.95 |
| PRF | 51.52 | **41.65** | **46.58** |

| Input of PRF | | mAP$_{base}$ ↑ | mAP$_{novel}$ ↑ | mAP$_{all}$ ↑ |
|---|---|---|---|---|
| box coord | point cloud | | | |
| × | × | 50.58 | 37.44 | 44.01 |
| ✓ | × | 49.73 | 23.02 | 36.38 |
| × | ✓ | 45.43 | 8.64 | 27.04 |
| ✓ | ✓ | 51.52 | **41.65** | **46.58** |

Table 3: **Effectiveness of pesudo label refinement.** We analyze whether the removal of the pseudo label refinement module (PRF) affects weakly incremental learning performance on the SUN RGB-D dataset.

Table 4: **Ablation experiments on PRF.** The model achieves the best results only when both the positional information of the bounding box and the contextual information from the point cloud are taken into account.

| Method | Vanilla | | | Ours | | |
|---|---|---|---|---|---|---|
| | mAP$_{base}$ ↑ | mAP$_{novel}$ ↑ | mAP$_{all}$ ↑ | mAP$_{base}$ ↑ | mAP$_{novel}$ ↑ | mAP$_{all}$ ↑ |
| Faster RCNN | 47.72 | 24.82 | 36.27 | 50.63 | 37.89 | 44.26 |
| Grounding Dino | 47.78 | 32.16 | 39.97 | 51.52 | 41.65 | 46.58 |
| 2D-Oracle | 50.38 | 34.58 | 42.48 | 52.58 | 46.10 | 49.34 |

Table 5: **Robustness to different 2D teachers.** We organize ablation studies to validate the robustness of our method to different 2D teachers. "Vanilla" denotes the baseline without PLR (details in Sec. 3.3), CKT and RKD (details in Sec. 3.4).

**Robustness to different 2D teachers.** To validate the robustness of our approach across different 2D teachers, we employed three distinct 2D teachers in our framework: "Faster R-CNN (Girshick, 2015)", "Ground Dino (Liu et al., 2023a)", and 2D human annotations ("2D Oracle"). Specifically, we train "Faster R-CNN" using 2D box annotations from the SUN RGB-D dataset (Song et al., 2015), and we directly infer "Ground Dino" on the image from SUN RGB-D dataset without any fine-tuning. "2D Oracle" represents the results annotated by human experts on these images. The results in Tab. 5 demonstrate improvements achieved by our approach with each of the 2D teachers. Our method not only improves the performance of existing detectors, such as achieving a $+3.74\%$ improvement on the base classes and a $+9.49\%$ improvement on the new classes for Ground Dino, but it also demonstrates a significant enhancement of nearly $+7\%$ across all categories when applied to manually annotated 2D results, which makes training 3D network using 2D prompts a viable possibility.

**Effectiveness of Bipartite Cross-modal Knowledge Transfer.** In Tab. 6, we compare our proposed **C**ross-modal **K**nowledge **T**ransfer (CKT) strategy with the "one-to-many" assignment strategy built within VoteNet, and the baseline method without $\mathcal{L}_{ckt}$. One can see that the "one-to-many" strategy performs even worse than the baseline method without the feature-level supervision ($-0.35\%$ mAP$_{novel}$), which is because of the noisy regional representations caused by occlusion of objects in 2D images. Meanwhile, our proposed CKT is able to help $\mathcal{S}^{3D}$ learn robust novel knowledge representations ($+1.43\%$ mAP$_{novel}$).

**Comparison with Other Knowledge Distillation Strategies.** We conduct experiments in Tab. 7 to compare the effectiveness of our proposed **R**eweighting **K**nowledge **D**istillation (RKD) strategy with other commonly used knowledge distillation strategies. To be specific, (Hinton et al., 2015) computes the **K**ullback-**L**eibler (KL) divergence, while (Zhao & Lee, 2022) computes the $l_2$ distance of the semantic logits for each proposal between the teacher and student model. As shown in Tab. 7, our proposed RKD achieves a higher performance for both base (51.52 mAP$_{base}$) and novel (41.65 mAP$_{novel}$) classes.

| Matching Strategy | mAP$_{base}$ ↑ | mAP$_{novel}$ ↑ | mAP$_{all}$ ↑ |
|---|---|---|---|
| - | 51.28 | 40.22 | 45.75 |
| One-to-Many | 50.76 | 39.87 | 45.32 |
| One-to-One (ours) | 51.52 | **41.65** | **46.58** |

| Distillation | mAP$_{base}$ ↑ | mAP$_{novel}$ ↑ | mAP$_{all}$ ↑ |
|---|---|---|---|
| - | 49.97 | 38.40 | 44.18 |
| Hinton et. al. | 51.13 | 39.14 | 45.13 |
| Zhao et. al. | 50.41 | 41.11 | 45.76 |
| RKD (ours) | **51.52** | **41.65** | **46.58** |

Table 6: **The performance of cross-modal knowledge transfer (CKT) by bipartite matching.** We compare the CKT utilizing bipartite matching with the unmatched approach. "-" denotes the absence of CKT.

Table 7: **Effectiveness of reweighting knowledge distillation (RKD) for weakly incremental 3D object detection.** We compare our proposed RKD with other commonly used knowledge distillation manner. "-" denotes that no distillation technology is used.

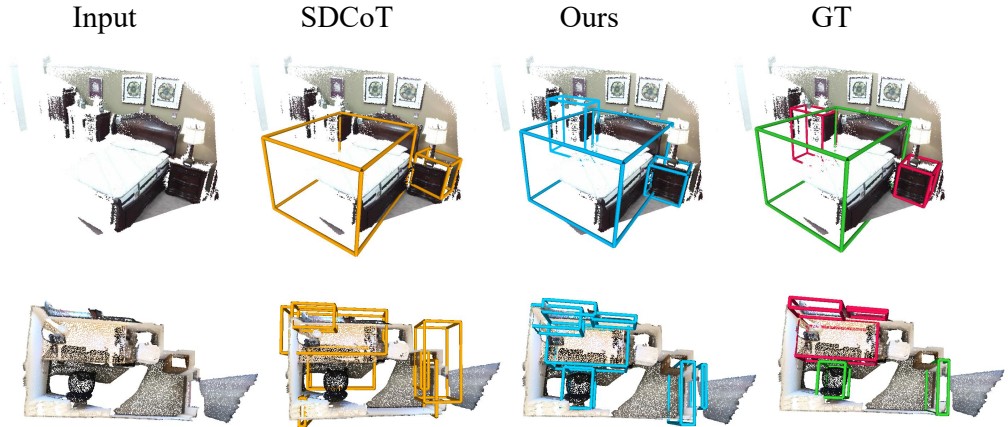

Figure 5: **Visualization of detection results.** Our proposed method is able to generate tight bounding boxes for both novel classes and base classes in these complex and diverse scenes. 3D ground truth annotations for these scenes are marked for base and novel classes repectively.

## 4.4 QUALITATIVE RESULTS

We showcase some qualitative results of our proposed methods on SUN RGB-D (Song et al., 2015) and ScanNet (Dai et al., 2017) in Fig. 5. One can see that our proposed method is capable of generating tight bounding boxes for both novel and base classes.

## 5 CONCLUSIONS AND LIMITATIONS

In this paper, for the *first* time, we attempt to address **W**eakly **I**ncremental **3**D object **D**etection, dubbed WI3D, which is a new task to study how to introduce both the **continous** *localization* and *recognition* ability of novel classes to a 3D detector through cost-effective 2D visual prompts. By learning from both inter- and intra- modal teachers, we propose (1) the pseudo label denoising technology to improve the quality of noisy 3D pseudo labels generated from visual prompts, and (2) concept representation learning in feature space for both base and novel classes. Experiments on SUN-RGBD and ScanNet validate that our proposed framework surpasses all baselines, including the previous approach to class-incremental 3D object detection. However, our method has the following limitations: 1) the proposed framework currently can not handle the novel categories that are not included in the vocabulary of 2D models. 2) there is still a gap between our results and those obtained using 3D annotations for novel classes. We leave them for future works and we fervently aspire that our endeavors in the realm of label-efficient 3D class-incremental learning tasks will spark inspiration and fuel future explorations in this community.

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
