# Supplementary Material for WI3D: Weakly Incremental 3D Detection via Visual Prompts

## A Implementation Details

**Details on Modified 3D Backbone.** Following SDCoT(Zhao & Lee, 2022), we adopt the modified VoteNet(Qi et al., 2019) as the detection backbone for the 3D teacher $\mathcal{T}^{3D}$ and student model $\mathcal{S}^{3D}$. Here, i) all indices of sampled points and votes from the student model are shared with the 3D teacher, ii) the prediction head is decoupled into the category classifier and class-agnostic localizer.

**Details on Coarse Pseudo Labels Generation.** Despite the one-to-one point-to-pixel correspondence between the input point cloud and its 2D image, it is hard to directly estimate a tight 3D bounding box based on a 2D one. Thus, we adopt a *cluster-then-refine* strategy in the way similar to that of (Peng et al., 2022), and the pseudo label refinement module is introduced in the main paper. The clustering algorithm first projects the point cloud onto the image plane and selects points within each 2D bounding box. Then, we adopt DBSCAN (Ester et al., 1996) to divide the points in each 2D box into multiple clusters according to the distance and density between the points. After that, we drop the clusters which contain fewer points than $1/10$ of the number of points in that 2D box, which ensures the generation of a tighter 3D instance mask. Finally, this process selects the cluster with the largest population, and calculates a coarse 3D bounding box with PCA to estimate the rotation, center, and size of the 3D pseudo box. Additionally, a simple cluster method cannot precisely distinguish the targets from noisy points. Therefore, we propose a class-agnostic Pseudo Label Refinement ReFinement (PRF) module to estimate more accurate 3D box pseudo labels based on the 3D pseudo boxes calculated from the above clustering algorithm for novel-class objects.

**Training and Inference.** In complementary to the framework overview stated in the main paper, we summarize the training process of our proposed framework for WI3D. In base train stage , we train a 3D teacher on base classes under the supervision of base-class 3D annotations. In Stage 2, we train a 3D student that learns novel-class concepts from a pretrained 2D teacher in a class-incremental manner and avoids forgetting previous knowledge through a 3D teacher without relying on continual 3d annotations. During the inference phase, we only need a point cloud as the input of the well-trained student network to detect all object categories it has learned.

## B Future analysis

To verify the effectiveness of our PRF, we adopt the box IoU with ground truth as a criterion to compare coarse labels $\hat{b}_{i,j}^{3D}$ and refined labels $\tilde{b}_{i,j}^{3D}$ on SUN-RGBD (Song et al., 2015). Considering that PRF only takes effect in the second stage of training, we conduct the experiments on the training set. The results are shown in Figure 6, and with the same scenes and 2D images, the box IoU of $\tilde{b}_{i,j}^{3D}$ remarkably surpasses the coarse $\hat{b}_{i,j}^{3D}$(e.g., +14.81 in table), which demonstrates that PRF can significantly render the estimated bounding box close to ground truth.

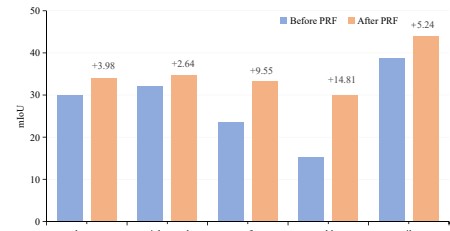

Figure 6: **Box IoU before and after PRF**. PRF can significantly improve the box IoU of each class, which implies outstanding denoising capacity of PRF.

# C VISUALIZATIONS

**More Examples** Here, we showcases more visualizations from our proposed approach on SUN RGB-D(Song et al., 2015) validation set. As illustrated in Fig. 7 and Fig. 8, our method demonstrates the capability of recognizing objects of both base and novel classes in a 3D scene.

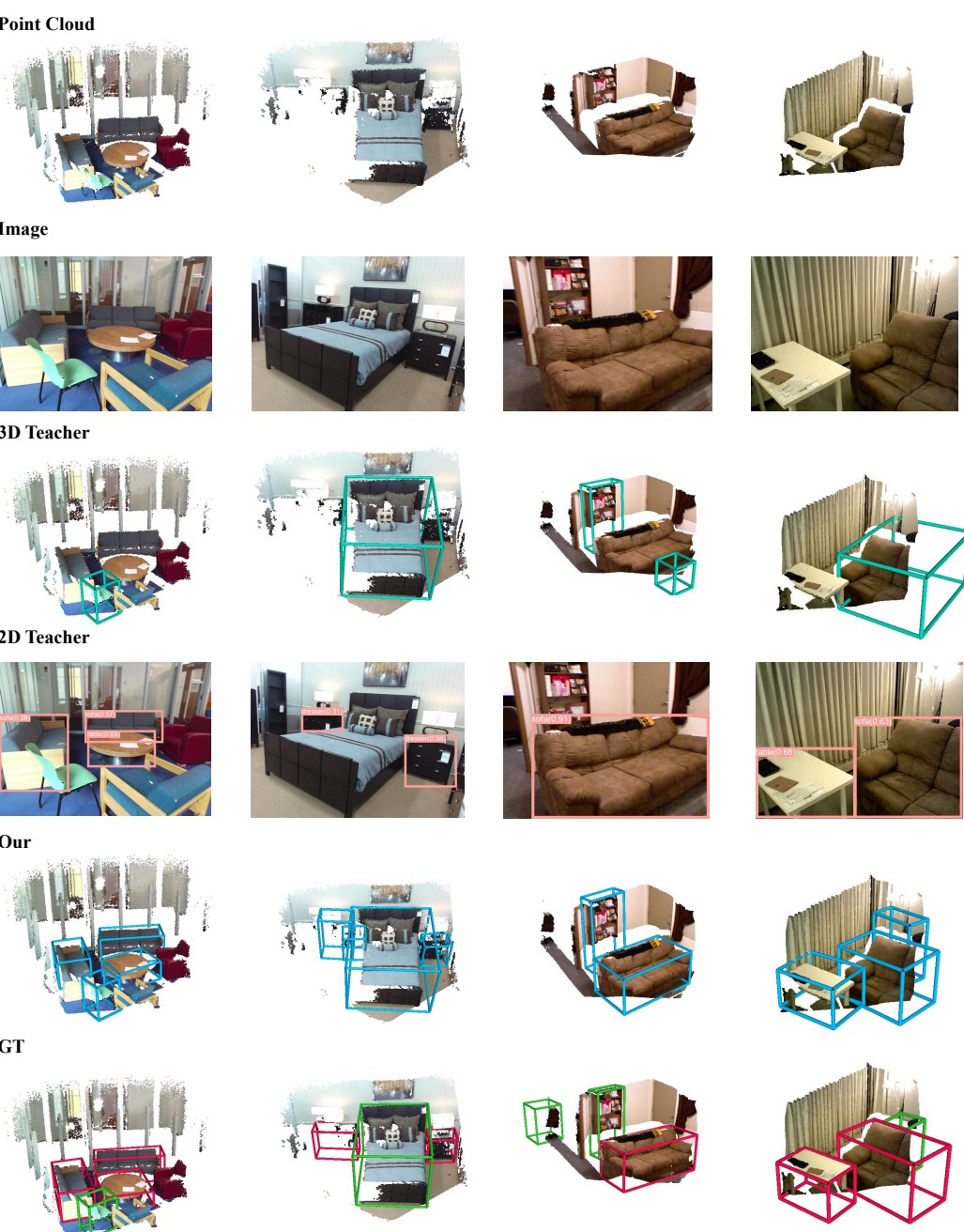

Figure 7: **More qualitative visualization results on SUN RGB-D validation set.** Green and red represent ground-truth annotations of base and novel classes, respectively. Best viewed in color with zoom in.

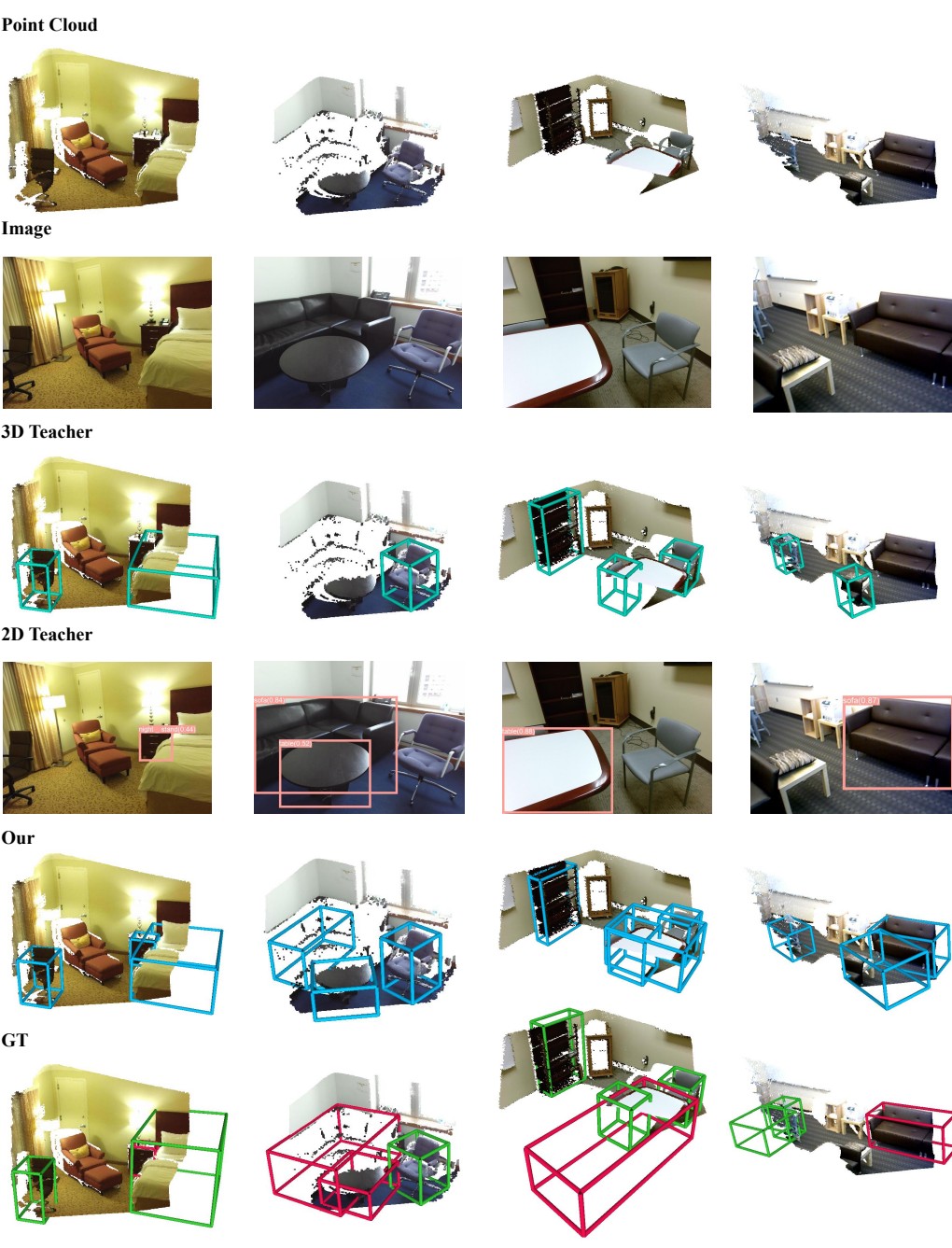

Figure 8: **More qualitative visualization results on SUN RGB-D validation set.** Green and red represent ground-truth annotations of base and novel classes, respectively. Best viewed in color with zoom in.

**Mipartite Matching Results** In Sec.3.4, we represent cross-modal feature alignment as a bipartite matching problem. Specifically, to associate each object in the image with its corresponding object in 3D space, we project the predicted 3D bounding boxes onto the image plane using camera parameters. Then we calculate the maximum and minimum coordinates to form the projected bounding boxes on image plane. We represent the 2D predictions and projected 3D proposals as two distinct sets. For facilitating the optimal matching between these two sets—finding a corresponding 3D proposal for

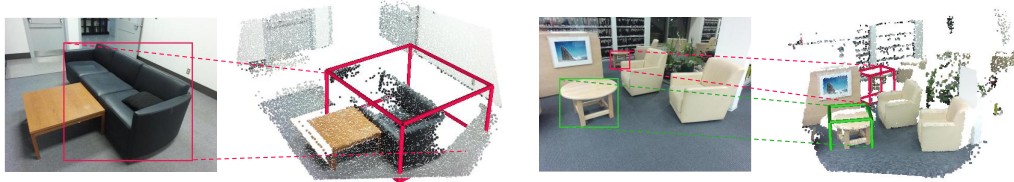

Figure 9: **2D-3D bounding box pairs by bipartite matching.** Objects enclosed within bounding boxes of the same color indicate paired items. Best viewed in color with zoom in.

each 2D prediction—we employ a bipartite matching method. The results of the 2D-3D matching can be observed in Fig. 9.