# OpenReview forum: "WI3D: Weakly Incremental 3D Detection via Visual Prompts"
_ICLR.cc/2024/Conference — ICLR 2024 Conference Withdrawn Submission_

### Official Review · Reviewer_h7RY · 2023-10-28

**Soundness:** 3 good
**Presentation:** 3 good
**Contribution:** 3 good
**Rating:** 6
**Confidence:** 3

**Summary:**

This paper proposed a framework WI3D, using cost-effective 2D visual prompts for weakly class-incremental 3D object detection, which is an unexplored but important field. Under the supervision of intra- and inter- modal teacher in both feature space and output space, WI3D could effectively learn the novel classes while retaining the knowledge of base classes. Experiments conducted on SUN RGB-D and ScanNet show that WI3D outperforms all other methods in weakly-supervised manner.

**Strengths:**

1. The topic of weakly incremental 3D detection is quite novel.
2. The paper makes clear illustrations of the major challenges and problems that need to be solved and proposes effective solutions correspondingly. And weaknesses of the proposed method are also clearly illustrated.
3. The proposed method is simple and easy to implement.
4. WI3D could reach relatively better performance. The extensive ablation study and analysis demonstrate the effectiveness of the proposed components.

**Weaknesses:**

1. There are two concerns about the method: (1) While generating the coarse pseudo labels, the method proposed in WEAKM3D is used. However, WEAKM3D is designed for the outdoor dataset, which is much sparser and instances are separate. It might lead to some failures and lower the quality of refined pseudo labels. (2) The knowledge transfer is only conducted on novel classes, while base classes still keep the original representation. It might confuse the model to learn quite different representations between novel classes and base classes.
2. Most technical parts can be found in other works and the technical contribution is marginal.
3. Some illustrations are not clear, e.g., how does teacher predict reweighting modulation factor; the “fine-tuning” method in 4.2 is described as “tune the whole model on novel classes”, while in I3DOD (Liang et al., 2023) and SDCoT (Zhao & Lee, 2022), it was described as “tune all parameters (except the old classifier) with a new classifier for C_novel”, I am not sure whether they are the same; what does vanilla method stands for in Table 5.
4. The quality of pseudo labels is of great importance. It would be better to show the mAP or recall of refined pseudo labels for further analysis, which is also a more intuitional way to clarify the effectiveness of PRF module.
5. The number of stages and total classes is a little small. I wonder how the method will scale under more learning stages and more classes (e.g. 10 stages for ScanNet200).

Wenqi Liang, Gan Sun, Chenxi Liu, Jiahua Dong, and Kangru Wang. I3dod: Towards incremental 3D object detection via prompting. arXiv preprint arXiv:2308.12512, 2023.
Na Zhao and Gim Hee Lee. Static-dynamic co-teaching for class-incremental 3D object detection. In Proceedings of the AAAI Conference on Artificial Intelligence, volume 36, pp. 3436–3445, 2022.

**Questions:**

1. “Weakm3d: Towards weakly supervised monocular 3d object detection.” was published in 2022 but not in 2021 as mentioned in the reference. You need to check the references carefully.
2. It would be better to show the performance of WI3D on outdoor datasets as supplementary experiments, for these are widely used for some promising areas like autonomous driving.

---

> ### Author Response · Authors · 2023-11-23
>
> We appreciate your approval of our idea and the insightful comments. Your concerns will be addressed in the following comments. We have also updated the revision of our paper accordingly.
>
> **[To weakness1, "generate the coarse pseudo labels"]**
> In our method, the 3D pseudo-labels are constructed in a way similar to that of Peng et al. (2021) [R1].To be specific, we first project the point cloud onto the image plane and select points within each 2D bounding box. Then, we adopt DBSCAN[R1](eps=0.25, min_samples=100) to divide the points in each 2D box into multiple clusters according to the distance and density between the points. After that, we drop the clusters which contain fewer points than 1/10 of the number of points in that 2D box, which ensures the generation of a 3D instance mask. In addition, we select the cluster with the largest cardinality and calculate a coarse 3D bounding box with PCA to estimate the rotation, center, and size of the 3D pseudo box.
> As this is not the primary focus of our innovation, we did not elaborate on it in the main paper. We have updated the details in the supplementary materials.
>
> **[To weakness1, "knowledge transfer"]**
> In section 3.4, we utilized an MLP-based projection layer to encode the 3D proposal features into the same feature space as the 2D region embedding. Similar to the representation of base-class objects, the representation of novel-class objects also receives direct supervision from pseudo labels. Additionally, prior works such as [R2] also propose knowledge transfer for novel-class objects for better learning of novel objects.
>
> **[To weakness2, "contribution"]**
> **Our key contribution is that we first propose and explore the task Weakly Incremental 3D object Detection**, which is an incremental 3D learning scheme in label-efficient scenarios. Because of the lack of novel-class 3D annotations and domain differences between images and point clouds, WI3D is a challenge and valuable task. To this end, we propose a **robust and effective** framework to deal with it, which contains a **class-agnostic pseudo-label refinement module** for high-quality pseudo-label generation and **concept representation learning** in feature space. Compared with the traditional workflow of 3D object detection, WI3D can further generalize the class-incremental 3D object detection to more realistic settings via cost-effective 2D visual prompts only. We earnestly aspire that our endeavors in the realm of weakly 3D class-incremental learning tasks will spark inspiration and accelerate future research in this community.
>
> **[To weakness3, "how does teacher predict reweighting modulation factor"]**
> During base training, we predict an objectness score[R3], indicating the predicted confidence that the object belongs to the base classes. While continuously learning novel classes, we utilize the softmax function to compute the objectness score of the current proposal concerning the total confidence among all proposals, acting as the reweighting modulation factor. We have updated the formula for calculating the reweighting modulation factor in the manuscript.
>
> **[To weakness3, "fine-tuning” method]**
> We apologize for the lack of clarity on this method. You're correct - we perform fine-tuning on the whole model(except the base classifier) and a new classifier for the Cnovel. We revised the description of the "fine-tuning" method in the manuscript.
>
> **[To weakness3, "vanilla method" in Table 5]**
> In Table 5, "vanilla" denotes the baseline without PLR (details in sec. 3.3), CKT and RKD (details in sec. 3.4). "Our" refers to the results achieved by our pipeline shown in Figure 3. We included the description for “vanilla” in the revised paper.
>
> **[To weakness4, "effectiveness of PRF module"]**
> Thank you for your valuable suggestions, we have provided results and analysis based on recall of coarse and refined pseudo labels in the supplementary material. The results demonstrate that PRF significantly improves the estimated bounding boxes of novel classes, bringing them closer to the ground truth labels.
>
> **Reference**
>
> [R1] Peng L, Yan S, Wu B, et al. Weakm3d: Towards weakly supervised monocular 3d object detection[J]. arXiv preprint arXiv:2203.08332, 2022.
>
> [R2]Zhao Z, Xu M, Qian P, et al. DA-CIL: Towards Domain Adaptive Class-Incremental 3D Object Detection[J]. arXiv preprint arXiv:2212.02057, 2022.
>
> [R3] Gupta A, Narayan S, Joseph K J, et al. Ow-detr: Open-world detection transformer[C]//Proceedings of the IEEE/CVF Conference on Computer Vision and Pattern Recognition. 2022: 9235-9244.

---

> ### Author Response · Authors · 2023-11-23
>
> **[To weakness5, "more learning stages and more classes"]**
> Thank you for your valuable suggestion. Due to time constraints, we conduct experiments specifically involving continual learning across 3 stages on ScanNet [R8], covering a total of 30 categories (in alphabetical order [R4]). It is evident that compared to SDCoT, our proposed method exhibits reduced susceptibility to catastrophic forgetting during sequential incremental learning. These experiments collectively demonstrate the superiority and robustness of our approach.
>
> #### Table r1: Per-class performance (AP@0.25) of SDCoT and WI3D on ScanNet val set under the sequential incremental learning setting. B[1-10] denotes standard training on ten base classes. N[11-20] and N[21-30] denote  sequential incremental training on the novel classes.
>
> |      Method  |      Phrase    |Number of evaluated classes|  $\{mAP}_{all}$|
> | :------------:| :------------: |:-----:|:-----:|
> ||B[1-10]|10|57.44|
> |SDCoT|+N[11-20]|20|30.38|
> |Our|+N[11-20]|20| **34.71**|
> |SDCoT|+N[21-30]|30|25.67|
> |Our|+N[21-30]| 30|**30.84** |
>
> **[To question1, "check the references"]**
> Thank you for your correction. We have addressed this issue in the revised version.
>
> **[To question2, "experiment on outdoor datasets"]**
> Compared to outdoor datasets, indoor scenes typically encompass a more diverse range of object categories, making them well-suited for incremental learning tasks. Previous works on class-incremental object detection focus solely on indoor scene datasets[R4-R6]. As a result, we conduct extensive experiments on two widely used indoor 3D datasets, SUN RGBD[R7] and ScanNet[R8], and compare our approach to traditional class-incremental learning methods.
> Due to significant differences in point cloud processing, model architecture, and training specifics between methods applied to indoor and outdoor datasets, we are presently unable to provide experimental results on outdoor datasets. Exploring weak incremental object detection under outdoor scenarios remains an area for future investigation.
>
> **Reference**
>
> [R4]Zhao N, Lee G H. Static-dynamic co-teaching for class-incremental 3d object detection[C]//Proceedings of the AAAI Conference on Artificial Intelligence. 2022, 36(3): 3436-3445.
>
> [R5]Zhao Z, Xu M, Qian P, et al. DA-CIL: Towards Domain Adaptive Class-Incremental 3D Object Detection[J]. arXiv preprint arXiv:2212.02057, 2022.
>
> [R6]Liang W, Sun G, Liu C, et al. I3DOD: Towards Incremental 3D Object Detection via Prompting[J]. arXiv preprint arXiv:2308.12512, 2023.
>
> [R7]Song S, Lichtenberg S P, Xiao J. Sun rgb-d: A rgb-d scene understanding benchmark suite[C]//Proceedings of the IEEE conference on computer vision and pattern recognition. 2015: 567-576.
>
> [R8]Dai A, Chang A X, Savva M, et al. Scannet: Richly-annotated 3d reconstructions of indoor scenes[C]//Proceedings of the IEEE conference on computer vision and pattern recognition. 2017: 5828-5839.

---

### Official Review · Reviewer_asnb · 2023-10-29

**Soundness:** 2 fair
**Presentation:** 3 good
**Contribution:** 2 fair
**Rating:** 5
**Confidence:** 5

**Summary:**

This paper addresses a novel task, which involves substituting 3D annotations of novel classes in class-incremental 3D object detection with 2D supervisions from images, termed Weakly Incremental 3D Object Detection (WI3D). The authors present a framework that extends the class-incremental 3D object detection method SDCoT. They adopt an existing approach for weakly supervised monocular 3D object detection to generate coarse 3D pseudo labels for novel classes. To address the challenges posed by noisy pseudo labels, they introduce a Pseudo Label Refinement (PLR) module, which directly refines these labels. Additionally, the authors employ cross-modal knowledge transfer techniques to enhance the robustness of the learned representation. Furthermore, they modify the base knowledge distillation loss within SDCoT that is used to mitigate the catastrophic forgetting of base knowledge. The proposed method is evaluated on two benchmark indoor datasets, under the batch incremental 3D object detection setting as proposed in SDCoT.

**Strengths:**

1. The problem of Weakly Incremental 3D Object Detection addressed in this paper is indeed a valuable area of exploration as it holds the potential to reduce annotation requirements significantly.
2. The proposed framework seamlessly integrates several pre-existing techniques, forming a technically sound architecture. The rationale behind the incorporation of each module is well-founded.

**Weaknesses:**

**1. The paper could benefit from providing more detailed and clear explanations.**

Firstly, the method for generating coarse pseudo labels is not clearly presented. While the authors mention adopting a method from (Peng et al., 2021), the specifics of this method need further clarification. It's essential to note that the method in Peng 2021 is intended for outdoor Lidar point clouds, which have different characteristics from indoor point clouds. Moreover, the process for predicting 3D object boxes in Peng 2021 may not be straightforward and may involve model training. Therefore, the paper should provide more details on how these coarse pseudo labels are generated and how they adapt to the indoor point cloud setting.

Secondly, the architecture and training strategy of the PLR module lack detailed explanations. Key aspects, such as the normalization step in Fig. 4, require clarification. Additionally, the training process of the PLR module in Stage 1, whether it is trained alongside the 3D detection backbone or separately, is not clear.

Thirdly, when comparing with SDCoT, the authors mention they “modify the training of SDCoT (Zhao & Lee, 2022) to fit our weakly incremental learning setting”, but there lacks a specific description of this modification (e.g., how to obtain annotations for novel classes). Also, the details of the fine-tuning and freeze-and-add setup in the WI3D task need to be clarified.

Fourthly, the paper mentions splitting the category set into C_base and C_novel according to SDCoT, but the procedure is not identical to SDCoT (e.g., comparing Table 1 in this paper vs. Table 1 in SDCoT).

Lastly, the term "vanilla" in Table 5 needs clarification. The distinction between "vanilla" and "ours" is not made clear.

**2. While the overall framework is sound, the paper's technical contributions are subject to doubt.**

Firstly, the PLR module appears to share significant similarities with the BoxPC network in [REF1], which is also designed to refine pseudo labels for novel classes by predicting 3D bounding box residuals and binary probabilities.

Secondly, when designing the intra-modal base knowledge distillation, the authors argue that “previous work usually directly utilizes all the predicted responses and treat knowledge equally.” As such, the reweighting modulation factor alpha_i should be the major modification compared to SDCoT. However, the authors omit comparing it with the version that removes alpha in the ablation study (refer to Table 7). This omission makes it challenging to discern the contribution of this modification.

**3. The paper does not present results in the sequential incremental learning setting, which is a common evaluation setting in Class-Incremental Learning.**

**4. The paper would benefit from visualization**, particularly in the form of 2D-3D bounding box pairs (bipartite matching results). Additionally, the validity of the example shown in Fig 2(a) is in question. I suspect such a misalignment might only occur when the 2D bounding box is entirely incorrect. The authors should provide a more detailed explanation or real-world examples to support the claim made in this figure.

[REF1] Tang, Yew Siang, and Gim Hee Lee. "Transferable semi-supervised 3d object detection from rgb-d data." Proceedings of the IEEE/CVF International Conference on Computer Vision. 2019.

**Questions:**

Please refer to the comments in the weaknesses section.

---

> ### Author Response · Authors · 2023-11-23
>
> We appreciate your approval of our idea and the detailed and insightful comments. Your concerns will be addressed in the following comments and the revision of our paper has been updated accordingly.
>
> **[To weakness1, "the method for generating coarse pseudo"]**
> In our method, the 3D pseudo-labels are constructed in a way similar to that of Peng et al. (2021).To be specific, we first project the point cloud onto the image plane and select points within each 2D bounding box. Then, we adopt DBSCAN[R1](eps=0.25, min_samples=100) to divide the points in each 2D box into multiple clusters according to the distance and density between the points. After that, we drop the clusters which contain fewer points than 1/10 of the number of points in that 2D box, which ensures the generation of a 3D instance mask. In addition, we select the cluster with the largest cardinality and calculate a coarse 3D bounding box with PCA to estimate the rotation, center, and size of the 3D pseudo box.
> As this is not the primary focus of our innovation, we did not elaborate on it in the main paper. We have updated the details in the supplementary materials.
>
> **[To weakness1, "clarification on normalization step and training process of the PLR"]**
> During the normalization process, the coordinates of the point cloud are restricted to the output range within [0, 1]. Afterward, we normalize the center coordinates and dimensions of the pseudo boxes to fit within the corresponding range.
> Furthermore, the PLR module is trained alongside the 3D detection backbone in stage 1, due to their shared input (point cloud) and supervision (ground truth labels for base class).
>
> **[To weakness1, "modify the training of SDCoT to fit our weakly incremental learning setting"]**
> We adaptively modify SDCoT to fit our WI3D setting. Specifically, we introduce the process of coarse pseudo-label generation to SDCoT to obtain annotations for novel classes, instead of using novel-class ground truth. Other training processes are identical to SDCoT. For a fair comparison, all the training settings, e.g., learning rate, optimizer, batch size, etc., are the same for all experiments.
>
> **[To weakness1, "details of the fine-tuning and freeze-and-add setup"]**
> "Fine-tuning" refers to first train the model on base classes, followed by fine-tuning the whole model as well as a new classifier for $C_{novel}$ except the classification heads for $C_{base}$. "Freeze-and-Add" is first trained on base classes, followed by adding a new classification head and fine-tuning only the new heads for $C_{novel}$.
>
> **[To weakness1, "splitting the category set into C_base and C_novel according to SDCoT"]**
> Following SDCoT, we also take a subset of classes (such as 7 base classes / 3 novel classes, 5 base classes / 5 novel classes in table 1) in alphabetical order from each dataset as $C_{base}$ and treat the remaining as $C_{novel}$. Additionally, we compare baseline methods, SDCoT, and our method under the same class partitioning.
>
> **[To weakness1, "the term "vanilla" in Table 5 needs clarification. "]**
> In Table 5, "vanilla" denotes the our method without PLR (details in sec. 3.3), CKT and RKD (details in sec. 3.4). "Our" refers to the results achieved by our pipeline shown in Figure 3. We included the description for “vanilla” in the revised paper.
>
> **Reference**
>
> [R1] Ester M, Kriegel H P, Sander J, et al. A density-based algorithm for discovering clusters in large spatial databases with noise[C]//kdd. 1996, 96(34): 226-231.

---

> ### Author Response · Authors · 2023-11-23
>
> **[To weakness2, "similarities with the BoxPC"]**
> Though BoxPC also refines pseudo labels for novel classes, our PLR differs from BoxPC in motivation, model design and settings.
>
> **Motivation**: As outlined in sections 3.1 and 3.3, the process of projecting 2D bounding boxes into 3D space introduces several challenges, including “projection migration", "scale ambiguity", and "overlap". To address these issues, we introduce a class-agnostic Pseudo Label Refinement (PLR) module. By decoupling 3D boxes and learning pseudo-label offsets and scales, our approach aims to resolve the problems stemming from Projection Migration and Scale Ambiguity. Furthermore, an additional Binary Classification Header (BCH) is employed to mask duplicated estimations. The BoxPC network, however, focuses on transferring knowledge learned from strong classes to weak classes and does not analyze or address the arising noise.
>
> **Model Design**: **1)** BoxPC manually designs specific thresholds such as bounds α+,α−,β+,β−. In contrast, PLR does not require manual threshold setting. **2)** BoxPC predicts a single fit probability p intended to constrain PointNet-based network training. In PLR, the Binary Classification Header (BCH) predicts the probability of presence compared to absence simultaneously, and the probability comparison helps mask duplicated estimations.
>
> **Settings**: **1)** BoxPC accepts only one 3D pseudo box and its corresponding local frustum point cloud as input. Conversely, multiple boxes and the whole scene are simultaneously fed in PLR, allowing it to utilize global and contextual information. **2)** During training, BoxPC utilizes pseudo-labels generated by perturbing 3D ground truth labels as input. However, PRF matches the disordered pseudo-labels generated from 2D visual prompts with the ground truth while refines the pseudo-labels.
>
> **[To weakness2, "comparing with removing alpha in the ablation study"]**
> We greatly appreciate your valuable suggestions. We supplement the corresponding experiments in Table r1, comparing our RKD with the results obtained by removing $\alpha$. The results in Table r1 demonstrate the effectiveness of our RKD with $\alpha$.
>
> #### Table r1: Effectiveness of reweighting knowledge distillation (RKD) for weakly incremental 3D object detection. We compare our proposed RKD with other commonly used knowledge distillation manner, as well as the method without $\alpha$. "-" denotes that no distillation technology is used.
> |      Method    |  $\{mAP}_{base}$ | $\{mAP}_{novel}$|  $\{mAP}_{all}$ |
> | :-------: |:-----:|:-----:|:-----:|
> |  -   |  49.97  |   38.40  |  44.18 |
> |  Hinton et. al. |  51.13| 39.14 |45.13 |
> | Zhao et. al. |   50.41 |41.11 |45.76  |
> |     RKD (w/o $\alpha$)     | 51.07   | 41.42  |  46.25 |
> |      RKD (ours)    |  **51.52** |**41.65** |**46.58**  |
>
> **[To weakness3, "results in the sequential incremental learning setting"]**
> We perform experiments under a sequential incremental learning scenario, where the novel classes are split into subsets and become available sequentially. We evaluate the per-class average precision (AP) when the novel classes are added in a sequential manner. The results are shown in Table r2. It can be observed that compared with SDCoT, the proposed method suffers less from catastrophically forgetting during sequential incremental learning. All those experiments demonstrate the superiority and robustness of our method.
>
> #### Table r2: Per-class performance (AP@0.25) of SDCoT and WI3D on SUN RGB-D val set under the sequential incremental learning setting. B[1-5] denotes standard training on five base classes. N[6-8] and N[9-10] denote sequential incremental training on the novel classes.
>
> |      Method  |      Phrase    |  bathtub| bed |bookshelf |chair |desk |dresser |nightstand |sofa| table |toilet | $\{mAP}_{all}$ |
> | :------------:| :------------: |:-----:|:-----:|:-----:|:-----:|:-----:|:-----:|:-----:|:-----:|:-----:|:-----:|:-----:|
> ||B[1-5]|75.16|84.52|32.47|74.82|25.23||||||58.54|
> |SDCoT|+N[6-8]|34.30|77.19|24.58|60.26|11.98|14.14|15.68|40.53|||34.83|
> |Our|+N[6-8]| 34.79 | 78.89 | 23.61 | 64.03 | 14.32 | 14.42 | 39.43 | 46.14 |||**39.45**|
> |SDCoT|+N[9-10]|25.20|74.58|18.92|53.26|8.79|11.71|3.79|40.29|12.93|67.00|31.65|
> |Our|+N[9-10]| 12.43 | 77.09 | 21.33 | 56.59 | 9.62 | 7.85 | 17.58 | 48.12 | 34.41 | 79.20 |**36.42**|
>
>
> **[To weakness4, "The paper would benefit from visualization and the validity of the example shown in Fig 2(a)"]**
> Thank you for your valuable suggestions and we have provided visual examples illustrating bipartite matching results in the updated supplementary materials. Furthermore, the illustration in column a of Figure 2 aims to elucidate the Projection Migration (Section 3.1 of the manuscript), which causes the displacement of the positions of the 3D bounding boxes. Additionally, we have replaced Figure 2 with a more illustrative one.

---

### Official Review · Reviewer_QdxM · 2023-11-01

**Soundness:** 3 good
**Presentation:** 2 fair
**Contribution:** 2 fair
**Rating:** 5
**Confidence:** 4

**Summary:**

This paper presents a method for class-incremental learning of 3D detection in RGBD pointcloud data. The method begins with a 3D detector trained on the base classes, and a 2D detector trained on all classes, and then attempts to transfer the novel-class knowledge from the 2D model to the 3D one. The method involves generating pseudo-labels using the 2D model, and then correcting these using the 3D model, and then training the 3D model with its own estimates. The method also incorporates two regularization losses, called the "Cross-Modal Knowledge Transfer" loss which increases cosine similarity between the 2D features and the 3D ones, and a "Reweighting Knowledge Distillation" loss that does a similar thing but (1) with a learnable weighting and (2) applied to logits too.

**Strengths:**

This paper is fairly well written, and does not make large over-claims about the novelty or impact or results. The figures are helpful in understanding the work. The method does well against its main considered baseline, SDCoT.

**Weaknesses:**

I have a variety of clarification questions that I hope the authors can address. I will put them into the Questions tab.

One clear weakness might be the evaluation. Why is there only one baseline in the evaluation? Other parts of the paper seem to acknowledge three closely related works: Zhao & Lee, 2022; Zhao et al., 2022; Liang et al., 2023. Is it possible to compare against all of them?

**Questions:**

The paper says "Inspired by the human visual system that excels at learning new 3D concepts through 2D images, we propose to incrementally introduce novel concepts to a 3D detector with the visual prompts generated from a cost-free 2D teacher other than revisiting 3D annotations for both base and novel classes as shown in Fig. 1." I do not understand the connection to the human visual system. I do not understand what it means for visual prompts to be "generated from a cost-free 2D teacher other than revisiting 3D annotations for both base and novel classes".

In Figure 2 it's unclear to me what method was used to generate the 3D boxes. The one in column b is especially egregious, since it seems like this does not even meet the edges of a plausible 2D box.

The method section says that it will "pose the noise of 3D pseudo labels directly generated from 2D predictions" and I don't know what this means. (What does it mean for noise to be posed or unposed?)

Section 3.1 says "we adopt a simple way to generate coarse 3D pseudo labels from 2D preditions" but it is never made clear what this method is.

Section 3.3 introduces a module called PRF which produces an offset to a given box. It is unclear how this module is trained. (Where do the ground-truth offsets come from? Does this training happen on all classes, or just base classes, or just novel classes?)

Section 3.3 introduces a module called BCH, which to my understanding takes the exact same input as the PRF module. If this is the case, it would be great to say so in the text, instead of re-stating the list of inputs as if it were unique. Also, it is unclear to me how this BCH module is trained. (Where does the ground truth "presence" label come from?)

Section 3.4 mentions difficulties associated with "extracting regional representations". This is fine, except that this is the first time in the method that regional representations are ever mentioned. What are they?

Section 3.4 mentions briefly that the IOU between the 2D predictions and the projected 3D predictions will be used "the cost function", but there is no equation given for this, and it's unclear to me if this is really one of the training objectives in Section 3.5. What is the exact form of the supervision? (Is it maybe the generalized differentiable IOU from Rezatofighi et al.?)




how to introduces -> how to introduce

Pseuod -> Pseudo

preditions -> predictions

objectiveness -> objectness

---

> ### Author Response · Authors · 2023-11-23
>
> Thank you for your valuable feedback. Your concerns will be addressed in the following comments and we have also updated the revision accordingly.
>
> **[To weakness, "only one baseline in the evaluation"]** The existing works include SDOCT (Zhao & Lee, 2022), DA-CIL (Zhao et al., 2022), and I3DOD (Liang et al., 2023). Among them, 1) Zhao & Lee (2022) introduce SDOCT for class-incremental 3D object detection which is open-sourced. In our evaluation, we assess SDCoT (Zhao & Lee, 2022) as the baseline model under the WI3D setting. 2) The official codebase of DA-CIL (Zhao et al., 2022) is currently **unavailable**. 3) Additionally, I3DOD (Liang et al., 2023), whose pre-print was initially released in August and was considered **concurrent work, does not have code available** to fit our settings.
>
> **[To question1, "the connection with the human visual system"]** As mentioned in [R1], humans can learn new visual concepts from a broad range of data modalities (cross-modal). In our case, we consider the 2D visual prompts as the cross-modal data. Perceiving a 3D environment through image is akin to viewing the world through human eyes, including constantly learning from images, and identifying corresponding objects in the 3D-physical world.
>
> **[To question1, "what it means for visual prompts to be"]** We utilize predictions (such as object‘s category, location, and features) generated by cost-free 2D teachers as "visual prompts". Unlike traditional class-incremental 3D object detection, we do not need delicate 3D annotations of novel-class objects, reducing the cost of 3D incremental learning.
>
> **[To question2 and question4, "the method used to generate the 3D boxes"]**
> In our method, the 3D pseudo-labels are constructed in a way similar to that of Peng et al. (2021).To be specific, we first project the point cloud onto the image plane and select points within each 2D bounding box. Then, we adopt DBSCAN[R2] to divide the points in each 2D box into multiple clusters according to the distance and density between the points. After that, we drop the clusters which contain fewer points than 1/10 of the number of points in that 2D box, which ensures the generation of a 3D instance mask. Finally, this process selects the cluster with the largest cardinality and calculates a coarse 3D bounding box with PCA to estimate the rotation, center, and size of the 3D pseudo box.
> As this is not the primary focus of our innovation, we did not elaborate on it in the main paper. We have updated the details in the supplementary materials.
>
> **[To question2, "the illustration in column b is especially egregious"]**
> Our method employs DBSCAN[R2] (a clustering technique) to eliminate noise caused by background points. Given the typically sparse nature of 3D point cloud surfaces, the 3D pseudo-labels might be smaller than actual objects. Consequently, this could lead to pseudo-labels not tightly covering the objects, as discussed in Section 3.1. Additionally, we have replaced Figure 2 with a more illustrative one.
>
> **[To question3,  posed or unposed"]**
> The probable noise we encounter is shown in Figure 2. The phrase "pose" might lead to mis-understanding, we have rephrased it with "present".
>
> **[To question5, "the training process of PRF"]**
> During base training(section 3.5), we train PRF only on base classes. The ground-truth offsets come from the difference between the annotations of the base class and the pseudo 3D labels. In the weakly incremental learning phase, we keep trained PRF frozen.
>
> **[To question6, "the input of BCH"]**
> As mentioned in Section 3.3, BCH serves as a classification head determining whether we retain the 3D pseudo label or not, aiming to address the issue of pseudo-label overlap. BCH takes distinct inputs compared to the PRF module: the input for PRF comprises the scene and coarse 3D pseudo-labels, producing refined pseudo-labels as output. In contrast, BCH takes output from the box encoder and context encoder as input, specifically the box-level features and scene context features.
>
> **[To question6, Where does the ground truth "presence" label come from?]**
> We use the Hungarian algorithm[R3] to match each annotation of the base class with an input pseudo label. The pseudo-labels assigned with ground-truth labels are treated as 1, and the rest are treated as 0. We have updated this part in our manuscript.
>
> **Reference**
>
> [R1]Bhunia A K, Gajjala V R, Koley S, et al. Doodle it yourself: Class incremental learning by drawing a few sketches[C]//Proceedings of the IEEE/CVF Conference on Computer Vision and Pattern Recognition. 2022: 2293-2302.
>
> [R2] Ester M, Kriegel H P, Sander J, et al. A density-based algorithm for discovering clusters in large spatial databases with noise[C]//kdd. 1996, 96(34): 226-231.
>
> [R3] Kuhn H W. The Hungarian method for the assignment problem[J]. Naval research logistics quarterly, 1955, 2(1‐2): 83-97.

---

> ### Author Response · Authors · 2023-11-23
>
> **[To question7, "regional representations"]**
> "regional representations" refer to the visual feature of a specific region in the 2D image extracted by a 2D teacher. We have updated the manuscript and specified its meaning.
>
> **[To question8, "the cost function"]**
> We represent the 2D predictions and projected 3D proposals as two distinct sets. To facilitate optimal matching between these two sets—finding a corresponding 3D proposal for each 2D prediction—the bipartite matching method necessitates a cost function, which can be formulated as follows:
>
>  $min \sum_i \sum_j m_{ij} * IoU(Project(B_{i}^{3D}),{B}_{j}^{2D})$
>
> $\sum_i m_{ij}=1$
>
> where $m_{ij} \in {0, 1}$ indicates whether it matches, and $IoU$ represents computing the intersection over union.
> We update the formula and explanation for the cost function in Section 3.4.
> Furthermore, the cost function mentioned above is not part of the training objectives.
>
> **[To question9, "introduce, Pseudo, predictions, objectness"]** Thank you for the notifications. We have fixed the typos accordingly.

---

### Meta-Review · Area_Chair_xirh · 2023-12-11

**Metareview:**

In this paper, the authors propose a label-efficient class-incremental 3D object detection method, which teaches a 3D detector to learn new object classes using cost-effective 2D visual prompts.

This paper received diverse ratings after the rebuttal period. One reviewer tended to accept this paper, while the other two reviewers decided to keep their original negative ratings. After reading the paper, rebuttal, and reviewers' comments, I tend to reject this paper for now. The reasons are as follows,

- There are some remaining concerns, e.g., the concerns mentioned by Reviewer asnb.
- No reviewers strongly support this paper.

The reviewers weren't actively involved in the author-reviewer discussion period. This makes it a little bit unfair for the authors. However, after checking this paper, I think its quality is not high enough for me to overturn the reviewers' decisions. I encourage the author to incorporate the reviewers' comments and resubmit the paper to the next venue.

**Justification For Why Not Higher Score:**

As I mentioned in my meta-review, there are still some remaining concerns, e.g., the concerns mentioned by Reviewer asnb. Besides, no reviewers strongly support this paper.

**Justification For Why Not Lower Score:**

N/A

---

### Decision · Program_Chairs · 2024-01-16

Reject